# Hypomethylation of *AHRR* (cg05575921) Is Related to Smoking Status in the Mexican Mestizo Population

**DOI:** 10.3390/genes12081276

**Published:** 2021-08-20

**Authors:** Omar Andrés Bravo-Gutiérrez, Ramcés Falfán-Valencia, Alejandra Ramírez-Venegas, Raúl H. Sansores, Rafael de Jesús Hernández-Zenteno, Andrea Hernández-Pérez, Leonor García-Gómez, Jennifer Osio-Echánove, Edgar Abarca-Rojano, Gloria Pérez-Rubio

**Affiliations:** 1HLA Laboratory, Instituto Nacional de Enfermedades Respiratorias Ismael Cosío Villegas, Mexico City 14080, Mexico; a.bravo.gtz@gmail.com (O.A.B.-G.); rfalfanv@iner.gob.mx (R.F.-V.); 2Tobacco Smoking and COPD Research Department, Instituto Nacional de Enfermedades Respiratorias Ismael Cosío Villegas, Mexico City 14080, Mexico; aleravas@hotmail.com (A.R.-V.); rafherzen@yahoo.com.mx (R.d.J.H.-Z.); andrea.hernandez@iner.gob.mx (A.H.-P.); leonor_garciag@hotmail.com (L.G.-G.); jennikkisx@hotmail.com (J.O.-E.); 3Clínica de Enfermedades Respiratorias, Fundación Médica Sur, Mexico City 14080, Mexico; raulsansores@yahoo.com.mx; 4Sección de Estudios de Posgrado e Investigación, Escuela Superior de Medicina, Instituto Politécnico Nacional, Mexico City 07340, Mexico; rojanoe@yahoo.com

**Keywords:** DNA methylation, cg05575921, cg23771366, *AHRR*, *PRSS23*, tobacco smoking

## Abstract

Tobacco smoking results in a multifactorial disease involving environmental and genetic factors; epigenome-wide association studies (EWAS) show changes in DNA methylation levels due to cigarette consumption, partially reversible upon tobacco smoking cessation. Therefore, methylation levels could predict smoking status. This study aimed to evaluate the DNA methylation level of cg05575921 (*AHRR*) and cg23771366 (*PRSS23*) and their correlation with lung function variables, cigarette consumption, and nicotine addiction in the Mexican smoking population. We included 114 non-smokers (NS) and 102 current tobacco smokers (TS); we then further subclassified them as heavy smokers (HS) (*n* = 53) and light smokers (LS) (*n* = 49). We used restriction enzymes (MspI/HpaII) and qPCR to determine the DNA methylation level. We observed significant hypomethylation of cg05575921 in smokers compared to NS (*p* = 0.003); further analysis found a difference between HS and NS (*p* = 0.02). We did not observe differences between other groups or a positive correlation between methylation levels and age, BMI, cigarette consumption, nicotine addiction, or lung function. In conclusion, the cg05575921 site of *AHRR* is significantly hypomethylated in Mexican smokers, especially in HS (≥20 cigarettes per day).

## 1. Introduction

Smoking kills more than 8 million people per year around the world. More than 7 million of these deaths are the result of direct tobacco use [1]. In Mexico, the latest national health survey reported that 17.9% of the adult population and 5.1% of people aged under 20 years are active smokers [2]. Tobacco smoke is a source of toxic substances such as carcinogens, toxins, fine particulate matter (PM), and addictive substances [3] that may be deposited in the human respiratory tract and cause disease [4].

Tobacco smoking results in a complex and multifactorial disease involving environmental and genetic factors. Studies in families and twins have shown that genetic factors contribute to the risk of smoking initiation, smoking persistence, nicotine dependence [5,6], smoking at a younger age [7,8], and others; previous reports mention that exposure to cigarette smoke has an important epigenomic link to changes in the genome [9,10]. These epigenetic mechanisms are highly dynamic and could contribute to chronic lung disease [11,12].

Multiple epigenome-wide association studies (EWAS) have found changes in DNA methylation levels due to tobacco smoking [13,14,15,16]. These studies have found CpG sites in distinct loci with low DNA methylation levels in independent studies and different populations; interestingly, three months after smoking cessation, this effect is partially reversible. The primary loci associated with DNA methylation in response to cigarette smoking are *PRSS23*, *RPS6KA2*, *GPR15*, *LRPS*, *CHRND*, *IER3*, and *AHRR* [17].

Previously, current smokers have shown hypermethylation of cg23771366 [17], located in the serine protease 23 (*PRSS23*) gene, which encodes a conserved member of the trypsin family of serine proteases [18]. Abnormal regulation of serine protease activity could lead to pathological conditions such as cancer [19]; in vitro assays have also proposed *PRSS23* as a potential biomarker of exposure to PM2.5 and reaffirmed its contribution to cancer development [18].

The hypomethylation of cg05575921 of the aryl-hydrocarbon receptor repressor (*AHRR*) gene has been consistently associated with smoking. Current smokers have lower DNA methylation levels and higher expression levels in this site than ex- and never-smokers in lung tissue [20]. The cg05575921 site is in an enhancer-like regulatory element within *AHRR*, which is a putative tumor suppressor gene whose expression downregulates the aryl hydrocarbon receptor (AHR); it is involved in pro-inflammatory signaling in human circulating monocytes [21,22] and is a critical regulator for metabolizing carcinogens from tobacco smoke, such as dioxin toxicity [23]. In addition, hypomethylation of cg05575921 in the blood may reflect an inflammatory response mediated by white blood cells [21].

The level of methylation of cg05575921 is associated with asthma development in infants born to smoking mothers [24], impaired lung function, COPD development and exacerbations [25,26], and carcinogenesis [22,23,26,27,28,29,30,31].

As such, hypomethylation at cg05575921 has been suggested as a potential biomarker for predicting smoking status [26] and daily cigarette consumption in saliva and blood in Caucasian and Afro-American populations [32,33]; however, this CpG site has not been evaluated in Latin American populations.

This study aimed to evaluate the DNA methylation levels of cg05575921 (*AHRR*) and cg23771366 (*PRSS23*) and their correlation with lung function variables, cigarette consumption, and nicotine addiction in the Mexican smoking population.

## 2. Materials and Methods

### 2.1. Study Population

The study was observational and retrospective, performed at the Instituto Nacional de Enfermedades Respiratorias Ismael Cosío Villegas (INER) in Mexico City; we selected samples of genomic DNA from the HLA Laboratory biobank. The participants were recruited from 2012 to 2019. The inclusion criteria for patient selection were as follows: Mexican mestizo ancestry (parents and grandparents born in Mexico), adults, non-smokers, or current smokers with normal lung function values. A specialized pulmonologist performed a physical examination and pre-bronchodilator spirometry, with reference values for the Mexican population used for the clinical assessment [19]. Individuals with exposure to biomass-burning smoke, other types of tobacco consumption and marijuana, occupational exposure including organic and inorganic dust, chemical agents and fumes, intensive air pollution (e.g., sculptors, gardeners, and warehouse workers), or diagnosed respiratory disease (COPD, bronchial asthma, bronchiectasis, active tuberculosis, lung cancer, cystic fibrosis, hypersensitivity pneumonitis, or idiopathic pulmonary fibrosis) were excluded from the study (Figure 1). The participants completed a questionnaire regarding demographical data and family history. In addition, a 7 mL sample of peripheral blood was collected in EDTA tubes. The individuals agreed to participate voluntarily and signed an informed consent document specifically for this protocol, previously approved by the INER research and biosafety bioethics committees (protocol code number B26-20).

### 2.2. Evaluation and Adjustment of the Concentration of Genomic DNA

The genomic DNA was evaluated using a NanoDrop 2000 spectrophotometer (Thermo Fisher Scientific, MA, USA); the absorbance measured the purity and concentration of the DNA at 260 nm and 280 nm. Our quality control reference points included samples with at least ≥150 ng/μL and a 260/280 ratio between 1.8 and 2.0. In addition, DNA integrity was evaluated in 2% agarose electrophoresis gels. All samples that met the above criteria were adjusted to 50 ng/μL for subsequent evaluation of methylation level.

### 2.3. Designing Primers against the Region Interest

We selected the CpG sites according to the previous reports in the literature of EWAS and tobacco smoking. These analyses were adjusted by age, sex, TI, BMI, and alcohol consumption according to a previous report that assessed them in four different populations [16]. We selected the 30 most significant candidates and then included only the sites with internal methylation in the CCGG tetranucleotide, resulting in four CpG sites.

We obtained the FASTA sequence [35] and verified that only one tetranucleotide 5′-CCGG-3′ sequence was present, which excluded two candidates; the final selection resulted in cg05575921 (*AHRR*) and cg23771366 (*PRSS23*). Next, we used the iMethyl database [36] to assess possible co-methylation influences [37] (Appendix A). Primers were designed in the Oligo Perfect tool (Thermo Fisher Scientific, Waltham, MA, USA) and evaluated in the UCSC in silico PCR tool [38].

### 2.4. Locus-Specific DNA Methylation

The EpiJET DNA Methylation Analysis Kit (Thermo Fisher Scientific, Waltham, MA, USA) was employed following the fabricant instructions. We used MspI and HpaII restriction enzymes (Thermo Fisher Scientific, Waltham, MA, USA) and primers described in Section 2.3 to analyze the DNA methylation status. The first part of the protocol involved incubating 0.5 μg of DNA at 37 °C for 120 min with three different reactions: undigested, MspI, and HpaII; the first was a control. Resultant fragments from such digestions can be analyzed using primers flanking the cleavage site; this part was performed with 2 μL of DNA, forward and reverse primers, and maxima SYBR green/ROX qPCR master mix 2× (Thermo Fisher Scientific, Waltham, MA, USA). We employed the StepOne system and the software v2.3 (Applied Biosystems, Waltham, MA, USA). We then selected 10% of our samples at random and duplicated our experiments (15 smokers and 15 non-smokers), including two negative template controls in each plate.

### 2.5. Data Analysis

Analysis was performed using R studio version 1.4.1106 [39] with the following libraries: tidyverse [40], ggpubr [41], rstatix [42], and nortest [43]. For each quantitative variable, the Kolmogorov–Smirnov test was performed to assess normality. As for sex, we performed a Fisher’s exact test in EpiInfo v7.0 [44]. We then performed a Spearman’s correlation and represented the results using the ggplot2 [45], hmisc [46], and corrplot [47] libraries. The statistical power was calculated using G-power 3.1.9.7 [48]. We adhered to the STREGA guidelines for reporting our results [34].

The qPCR efficiencies were determined using LinRegPCR provided by the Real-Time PCR Data Markup Language (RDML) consortium [49,50] (Appendix A for site in *AHRR* and S3 for site *PRSS23*). The average efficiency for *AHRR* was 94.2% and it was 97.9% for *PRSS23*. The percentage of methylation was calculated according to the following formula given in the manufacturer’s protocol:
% 5-mC = 1000/(1 + E) ^(Ct HpaII − Ct control)^(1)
where % 5-mC is the 5-methylcytosine percentage, E is the qPCR efficiency, Ct is the threshold cycle, Ct HpaII represents the Ct of “digested with HpaII,” and Ct is for an undigested DNA sample (Appendix A). The 5-mC percentage was evaluated in non-smokers (NS) and tobacco smokers (TS). Comparisons were performed with the Mann–Whitney U-test and a Bonferroni post-hoc test. The smokers were divided into heavy smokers (HS), with ≥20 cigarettes per day (cpd), and light smokers (LS) ≤10 cpd; comparisons were made with the Kruskal–Wallis test with a post-hoc Dunn test. A value of *p* < 0.05 was considered statistically significant in all our analyses.

## 3. Results

### 3.1. Subjects Included in the Study

The tobacco smokers’ group was predominantly male, with a lower body mass index (BMI), higher forced vital capacity (FVC), and lower FEV1/FVC than the non-smoker group. There were no significant differences in age (Table 1).

In addition, a comparison between tobacco smokers revealed that subjects in the HS group were predominantly male, had a higher addiction to nicotine as assessed by the Fagerström questionnaire, and had a higher tobacco index; their lung function showed no difference compared to LS (Table 2).

### 3.2. DNA Methylation Analysis

The methylation level in cg05575921 (*AHRR*) in the NS group had a median of 100% (interquartile range, IR = 26.96–100.0); meanwhile, in the TS group, the median was 47.16% (IR = 4.15–100.0). This is a significant difference, adjusted by post-hoc correction (Bonferroni) (*p* = 0.003, Figure 2a); these results have a statistical power of 92.7%. The cg23771366 site (*PRSS23*) did not show significant differences (*p* = 0.276, Figure 2b).

The 5-mC percentage between HS, LS, and NS for the cg05575921 site was minor (medians of 40.43%, 57.86%, and 100%, respectively) for higher cpd (*p* = 0.01); however, after post-hoc correction, there was only a difference between HS and NS (*p* = 0.02, Figure 3a). The cg23771366 site did not show statistical differences (Figure 3b).

### 3.3. Correlation Analysis in Tobacco Smokers’ Group

We performed a Spearman correlation analysis in the tobacco smokers’ group with cg05575921 (*AHRR*) or cg23771366 (*PRSS23*) and age, BMI, Fagerström test for nicotine dependence (FTND), tobacco index (TI), cpd, and pre-bronchodilator lung function. However, the methylation level in both sites evaluated had no significant correlation (Figure 4a,b).

## 4. Discussion

We observed and compared methylation levels in the cg05575921 (*AHRR*) and cg23771366 (*PRSS23*) sites in the Mexican mestizo population. To the best of our knowledge, this is the first study describing the methylation levels in Mexican smokers in CpG sites related to tobacco consumption. We observed a significant difference in the methylation level of the cg05575921 site, with hypomethylation in the tobacco smokers’ group compared to non-smoking subjects.

Smoking is a significant public health problem; 60.7% of current smokers in Mexico wish to quit, and 22.1% manage to stay abstinent for at least one year [51]. We observed that male sex was predominant in the TS group, while in NS, females constituted 80%. The TS had a lower BMI in comparison with NS. The FVC and FEV1/FVC were lower in the tobacco smokers’ group, without reaching lung obstruction. Stratification of tobacco smokers in HS and LS showed that HS were older than LS; moreover, the FTND punctuation and TI were higher than LS. However, LS had a lower BMI than HS.

The methylation levels of cg05575921 (*AHRR*) and cg23771366 (*PRSS23*) were evaluated in the whole blood of the tobacco smoking and non-smoking Mexican population. We found that site cg05575921 of TS had lower methylation levels than NS (*p* = 0.003) and there were differences between heavy smokers and non-smokers (*p* = 0.020).

The current study describes the methylation levels in Mexican non-smokers and smokers in *AHRR* (cg05575921). Previously, in the European American population of around 44 years and 17.9 pack-years, the mean methylation level reported in cg05575921 was 47.1%; meanwhile, younger African Americans (29 years of age and roughly 0.07 pack-years) presented 58.5% of methylation in cg05575921 [52]. Recently, in a group of white participants (54.2 pack-years), the mean methylation level was 49.2%; in this same study, the control population (non-smokers) showed 86.2% of methylation [32]. Our study identified similar levels (Figure 5); however, further investigations should be carried out considering the smokers’ age, smoking history, population ancestry, and topography of cigarette consumption.

It should be noted that the studies shown in Figure 5 employed bisulfite conversion to determine the methylation level, and in the present study, we used a technique without this conversion because it was previously reported that incomplete bisulfite conversion can introduce artifacts [53]; however, despite the different techniques used to assess the level of methylation, hypomethylation was found to exist only in subjects with cigarette consumption.

Currently, there are three methods for determining tobacco consumption. The first is the self-report; however, it is subject to bias as it relies on the memory of each patient [54]. The second is exhaled carbon monoxide levels; unfortunately, this is only helpful in detecting tobacco smoking within 3–4 h from the last cigarette smoked [55]. Finally, the most sensitive are cotinine serum or salivary levels; however, these can only detect cigarette consumption after up to 48 h [56]. Several specialists confirm that one of the most considerable barriers to the development of more effective smoking prevention and cessation interventions has been the relative inability to quantify tobacco consumption objectively [57].

In Caucasian and Afro-American populations, the methylation status at a CpG locus in the *AHRR*, cg05575921, is sensitive and specific for smoking status in adults, and there is reversibility in methylation levels up to 3 months after smoking cessation [57]. Therefore, this epigenetic assessment objectively ascertains smoking status. As a result, increasing research proposes evaluating the methylation status in CpG sites related to smoking as a biomarker of smoking status [25,26,52,55,56,58,59]. Understanding the relationship between the epigenetic effects of smoking can increase our knowledge of the damage mechanisms that lead to chronic diseases related to tobacco smoking [14,23,27,28,29,30,31,60,61].

Our methylation levels for cg05575921 were similar to previous reports realized in different populations with tobacco consumption [20,25,32,62,63] and non-smokers [25,63], showing replicable results in most populations studied; this is relevant because, previously, cg05575921 has been proposed as a biomarker of smoking status since DNA methylation is not affected by short-term abstinence periods [32].

It is of great importance to evaluate whether reversibility in methylation levels is observed in Latin American populations after an intervention to stop smoking and at three months, and to expand the studies to demonstrate that the cg05575921 can be a biomarker of smoker status.

Some studies have found negative correlations among cigarette consumption [7], leading us to believe that HS have lower methylation levels for cg05575921. Previous reports have concluded that this is a limitation, as intermittent smoking in light smokers is a frequent phenomenon, impairing these methods’ reliability for detection [32]. Therefore, evaluating cg05575921 as a biomarker of smoking status in Mexican patients undergoing treatment to quit smoking is necessary.

Regarding the methylation levels of cg23771366, we did not observe a significant difference between TS and NS compared to previous reports [23,64]. While cg23771366 is related to smoking, its most significant association is with lung function [17]; it is likely that, as our studied population had “healthy lung function,” differences between the methylation percentage in TS and NS were not observable.

We also performed a correlation analysis in which we observed that, while not significant, methylation levels had negative correlations with lung function, cpd, and FTND. For example, in the Caucasian population, it has been reported that *AHRR* hypomethylation was associated with low lung function, steeper lung function decline, and respiratory symptoms [63]; however, in a cohort from England, the authors reported negative evidence for *AHRR* methylation on FEV1, indicating that it is unlikely to be mediating the effect of smoking on lung function [65].

This research is not exempt from limitations; among these, we only had access to the available data in the clinical history obtained from the participants, which did not include the puffing topography and biochemical assessment of tobacco. Similar to other reports [32,52], our data show high dispersion in the methylation distribution; tobacco smoking is a complex process influenced by several mechanisms, including the intensity of consumption, which varies among populations [66]. Additional validation studies are necessary for other populations in order to confirm that cg05575921 could be a biomarker for predicting smoking status; in the Mexican population, it is necessary to realize the measurement of methylation levels in the same subjects before and after undergoing a program to quit smoking.

## 5. Conclusions

The cg05575921 site of *AHRR* is hypomethylated in smokers compared with non-smokers in the Mexican mestizo population; this hypomethylation is statistically significantly higher in heavy smokers than non-smokers.

## Figures and Tables

**Figure 1 genes-12-01276-f001:**
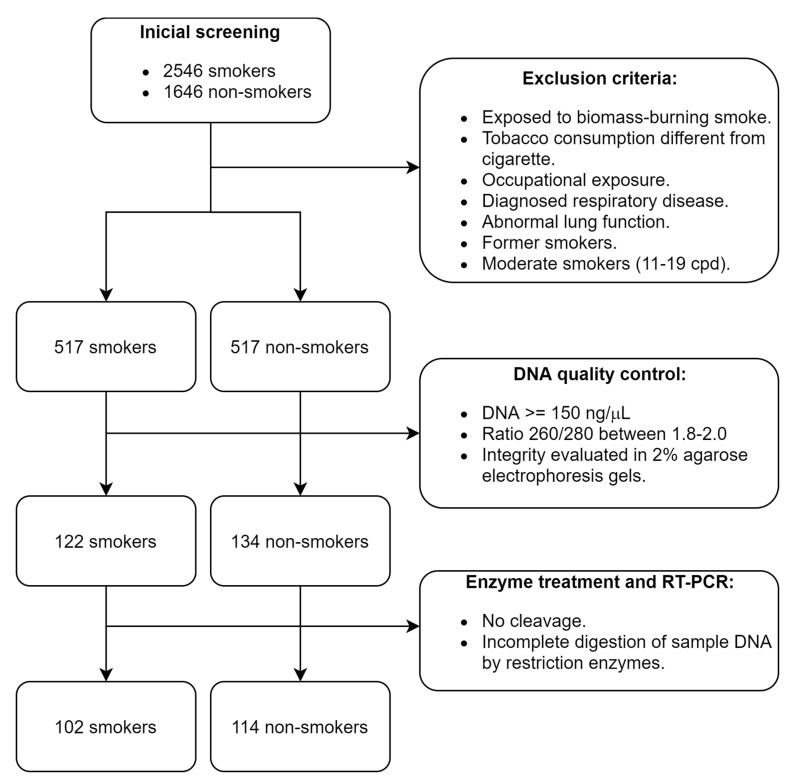
Patient selection process according to the STREGA guidelines [34].

**Figure 2 genes-12-01276-f002:**
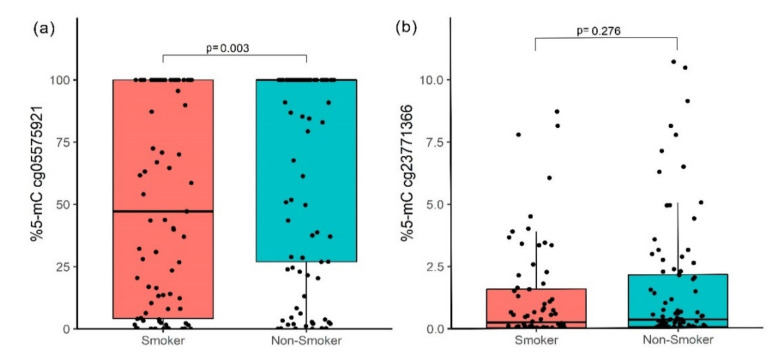
Methylation percentage between smokers and non-smokers. (**a**) cg05575921 (*AHRR*) site and (**b**) cg23771366 (*PRSS23*) site; the *p*-value was obtained by the Mann–Whitney U test with a Bonferroni post-hoc correction.

**Figure 3 genes-12-01276-f003:**
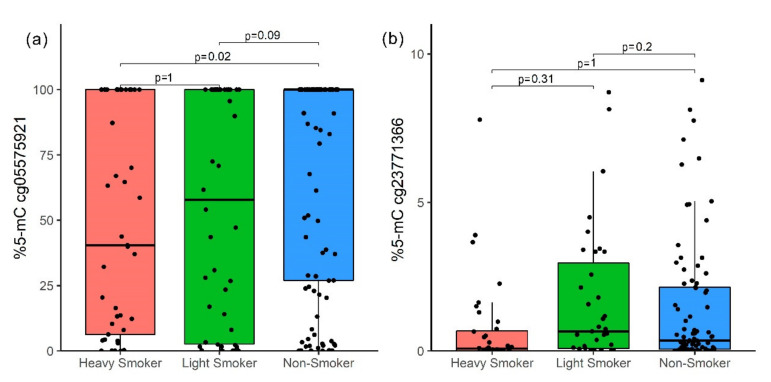
Methylation percentage comparison between HS, LS, and NS; *p*-value by Bonferroni correction. (**a**) cg05575921 (*AHRR*) site and (**b**) cg23771366 (*PRSS23*) site. The *p*-value was obtained by the Kruskal–Wallis test with a post-hoc Dunn test.

**Figure 4 genes-12-01276-f004:**
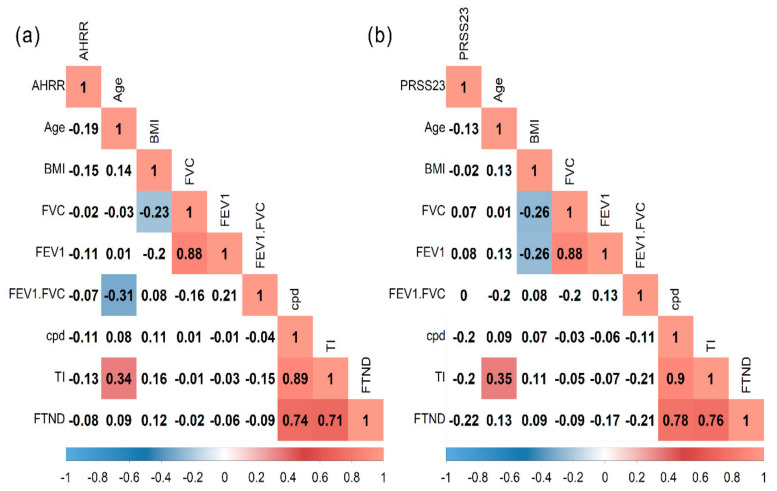
Spearman correlation matrix in tobacco smokers’ group. (**a**) cg05575921 (*AHRR*) site and (**b**) cg23771366 (*PRSS23*) site. BMI, body mass index; FVC, forced vital capacity; FEV1, forced expiratory volume in the first second; cpd, cigarette per day; TI, tobacco index; FTND, Fagerström test for nicotine dependence.

**Figure 5 genes-12-01276-f005:**
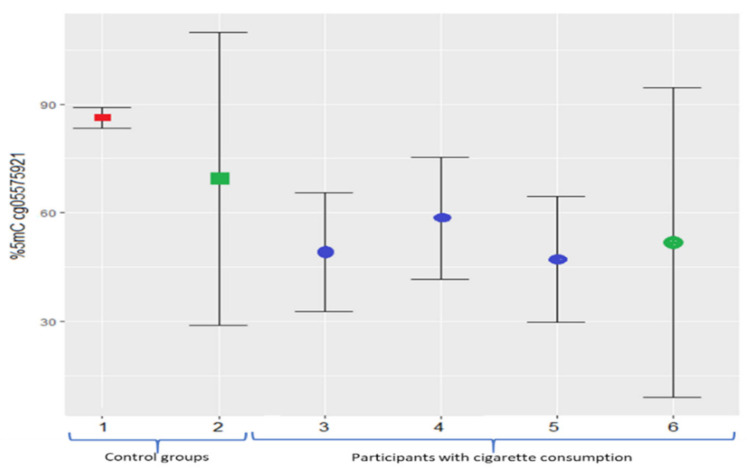
Mean methylation percentage (with standard deviation error bars) of cg05575921 (*AHRR*) site. The red square represents the non-smokers’ group; blue circles represent European American and African American smokers. The green square and circle represent the data obtained in this study. 1, non-smokers and white ethnicity [32]; 2, non-smokers in our study; 3, smokers with white ethnicity [32]; 4, African American smokers [52]; 5, European American smokers [52]; and 6, smokers included in this study.

**Table 1 genes-12-01276-t001:** Clinical and demographic overview of the studied groups.

Variable	Tobacco Smokers (*n* = 102)	Non-Smokers (*n* = 114)	*p*
Age (years)	47 (39−56)	50 (34−58)	0.4200
Sex (male %)	44.1	20.1	0.0002 *
BMI (kg/m^2^)	26.1 (23.9−28.9)	28.09 (24.6−32.0)	0.0030
Lung function pre-bronchodilator			
FVC (%)	96.8 (88−106)	91.0 (85−99)	0.0100
FEV1(%)	97 (87−105)	96 (86−105)	0.6000
FEV1/FVC (%)	81.2 (77−85)	85.0 (81−92)	1.5 × 10^−7^

BMI, body mass index; FVC, forced vital capacity; FEV1, forced expiratory volume in the first second. Results are shown in median and interquartile range except for sex *. The *p*-value was obtained by the Mann–Whitney U test.

**Table 2 genes-12-01276-t002:** Clinical and demographic overview of smokers.

Variable	Heavy Smokers (*n* = 53)	Light Smokers (*n* = 49)	*p*
Age (years)	50 (42−56)	43 (32−56)	0.0300
Sex (male %)	58.4	28.5	0.0040 *
BMI (kg/m^2^)	26.7 (24.5−28.9)	25.8 (23.2−28.9)	0.3100
FTND	6 (5−8)	1 (0−3)	1.1 × 10^–10^
Tobacco index (packs/year)	34 (26−45)	5 (1−10)	2.2 × 10^–16^
Lung function pre-bronchodilator			
FVC (%)	96 (89−104)	96 (88−107)	0.9800
FEV1 (%)	97 (87−105)	98 (89−97)	0.9800
FEV1/FVC (%)	80.2 (76−85)	82.0 (79−85)	0.2300

BMI, body mass index; FTND, Fagerström test for nicotine dependence; FVC, forced vital capacity; FEV1, forced expiratory volume in the first second. Results are presented in median and interquartile range except for sex *. The *p*-value was obtained by the Mann–Whitney U test.

## Data Availability

The data presented in this study are available in the Appendix A.

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
