# Peer review of "Hypomethylation of AHRR (cg05575921) Is Related to Smoking Status in the Mexican Mestizo Population"

_genes, 2021, doi:10.3390/genes12081276_

Round 1
Reviewer 1 Report
None
Author Response
Thank you for your comments. We attach the answer.

Reviewer 2 Report
In this revised manuscript, the aims and background are a little clearer, though it that could still be improved. A large part of the discussion should be in the introduction: we need to know what has already been done and what motivated the study, rather than general facts about smoking. As I understand it, the logic is:
- Why do we need to know smoking status? Knowing status might help cessation interventions etc.
- How can we determine smoking status based on epigenetics? Reversible methylation etc.
- Does this hold in the Mestizo population? Need to test if known sites with association to smoking status could work as biomarkers for this population.
It appears that publication 51 is pivotal: methylation status at one CpG, along with gender and age, can be used to predict smoking status. However the technique used there is very different (bisulfite + digital droplet PCR) and much more quantitative than the restriction enzyme-based technique used by the authors. These differences in methods, how it relates to the accuracy of measurement and the impact on the results should be discussed.
Since the larger goal is to evaluate cg05575921 as a biomarker (l264-266 in Discussion, which should be in introduction), we need a conclusion about this goal. It seems that despite significant differences in average methylation, the methylation estimate is far too variable to make an efficient prediction of smoking status. Therefore, with the current experimental design, cg05575921 would by itself not be a good marker for the Mestizo population.
I note that the model in publication 51 included gender, therefore dismissing the effect of gender on the methylation values in the current studies seems too quick.
In Figures 2 and 3, raw (or raw-er) data should be plotted in addition to the box plots. In the presence of non-normal data, box plots can hide a lot on the true underlying distributions.
Some editing for English is required, in particular in the new sections. When talking about methylation level (l. 221-232 notably), please specify which metric is used (mean, median, ...). Figure 5 title should be “Mean methylation percentage” rather than “Methylation percentage of the mean”. I believe the statistics terms may be Bonferroni “correction” rather than “test” and “Mann-Whitney U-test” rather than “U-Mann Whitney test”.
Author Response
Thank you for your comments. We attach the answer.

This manuscript is a resubmission of an earlier submission. The following is a list of the peer review reports and author responses from that submission.
Round 1
Reviewer 1 Report
In this manuscript, Bravo-Gutiérrez et al. show that there is a significant difference in methylation at one CpG residue between smokers and non-smokers in the Mexican mestizo population. There is large overlap in the distribution of methylation between these two populations, so smoking status cannot be predicted from this CpG alone. This appears to contradict the title “methylation levels… serve as an indicator”. The aims of the study are not defined in the abstract or introduction, only the protocol (test association between methylation and smoking status/lung function). From the conclusion, it seems that the aim could be to find biomarkers for smoking, but the study design is not appropriate for biomarker discovery.
Overall the study has a very limited scope (confirming known association of methylation and smoking at 2 CpG sites) and does not make clear what is achieved.
The test performed to compare DNA methylation levels between groups is not reported (presumably Mann-Whitney U-test?). Otherwise the data collection and analysis appear well done.
Reviewer 2 Report
In the article, the authors report a significant change at cg05575921 site of AHRR gene in Mexican smokers. The article is focused and well written with sufficient and relevant literature background. I would like to point out on few issues, which I believe would further improve the paper as follows:
- Firstly, the authors used a specific protocol involving restriction enzymes in order to evaluate the methylation status of two specific CpG sites. However, the enzymes MspI and HpaII may work with different efficency. Their data could be more robust if they could confirm their results with another technique, such as Methylation-Specific PCR (MSP), also including a lesser number of individuals.
- Considering that DNA methylation status may be influenced also by other near CpGs (see for example, Affinito O et al., Genomics, 2020) did the authors investigated also other eventual CpG sites located on the selected loci?
- Please, increase the resolution of the images (the labels of the x-axis are not legible in Figure 2 and Figure 3)